

# From Ground Photos to Aerial Insights: Automating Citizen Science Labeling for Tree Species Segmentation in UAV Images

Salim Soltani[1,2,3*], Lauren E. Gillespie[3,4,5], Moises Exposito-Alonso[3,4], Olga Ferlian[6,8], Nico Eisenhauer[6,8], Hannes Feilhauer[2,6,7], and Teja Kattenborn[1]

[1]Sensor-based Geoinformatics (geosense), University of Freiburg, Freiburg, Germany
[2]Remote Sensing Centre for Earth System Research (RSC4Earth), Leipzig University, Germany
[3]Department of Plant Biology, Carnegie Science, Stanford, California, USA
[4]Department of Integrative Biology, University of California, Berkeley, Berkeley, California, USA
[5]Department of Computer Science, Stanford University, Stanford, California, USA
[6]German Centre for Integrative Biodiversity Research (iDiv), Halle-Jena-Leipzig, Germany
[7]Helmholtz Centre for Environmental Research, Leipzig, Germany
[8]Institute of Biology, Leipzig University, Germany
[*] Corresponding author: salim.soltani@geosense.uni-freiburg.de

## Abstract

Spatially accurate information on plant species is essential for various biodiversity monitoring applications like vegetation monitoring. Unoccupied Aerial Vehicle (UAV)-based remote sensing combined with supervised Convolutional Neural Networks (CNNs)-based segmentation methods has enabled accurate segmentation of plant species. However, labeling training data for supervised CNN methods in vegetation monitoring is a resource-intensive task, particularly for large-scale remote sensing datasets. This study presents an automated workflow that integrates the Segment Anything Model (SAM) with Gradient-weighted Class Activation Mapping (Grad-CAM) to generate masks for citizen science plant photographs, reducing the efforts required for manual annotation. We evaluated the workflow by using the generated masks to train CNN-based segmentation models to segment 10 broadleaf tree species in UAV images. The results demonstrate that segmentation models can be trained directly using citizen science-sourced plant photographs, automating mask generation without the need for extensive manual labeling. Despite the inherent complexity of segmenting broadleaf tree species, the model achieved an overall acceptable performance. Towards efficiently monitoring vegetation dynamics across space and time, this study highlights the potential of integrating foundation models with citizen science data and remote sensing into automated vegetation mapping workflows, providing a scalable and cost-effective solution for biodiversity monitoring.

Keywords: Remote Sensing, Convolutional Neural Network, Citizen Science Data, Plant species, Transfer learning, Segment Anything Model, Gradient-weighted Class Activation Mapping, Automated Mask Generation.



## 1 Introduction

Spatially explicit, timely and accurate information on the distribution of plant species is essential for a wide range of applications, including mapping biodiversity, endangered or invasive species in conservation monitoring, weed detection in precision agriculture, and tree species assessments in forest management. Remote sensing images from drones, also known as unoccupied Aerial Vehicles (UAVs), have emerged as an effective source of information for mapping plant species (Sun et al., 2021; Maes and Steppe, 2019; Lopatin et al., 2019; Curnick et al., 2021; Wagner, 2021; Müllerová et al., 2023; Bouguettaya et al., 2022; Fassnacht et al., 2016). Through mosaicing aerial images, UAVs enable the creation of orthoimages that cover relatively large areas with very high spatial resolution in the centimeter to millimeter range. The spatial detail in such imagery can reveal distinct morphological features for plant species identification. These features include leaf shapes, flower structures, branching patterns, and canopy structure.

Supervised deep learning techniques, particularly convolutional neural networks (CNNs), can successfully be used to exploit these spatial patterns for automated plant species identification (Kattenborn et al., 2019a; Schiefer et al., 2020; Brodrick et al., 2019). Particularly, CNNs for semantic segmentations enable an assignment of each pixel of a UAV orthoimage to a plant species, enabling the mapping of the spatial distribution of plant species in unprecedented detail (Kattenborn et al., 2021b; Hoeser and Kuenzer, 2020)

However, a key challenge with using supervised CNNs for plant species mapping is the need for large amounts of training data (Kattenborn et al., 2021b; Galuszynski et al., 2022). Especially when neighboring plant species look similar, a large amount of training data is needed to allow the model to learn the subtle differences between these species (Kattenborn et al., 2021b; Schiefer et al., 2020). Traditionally, training data has been derived from field surveys or manual annotation of UAV images. Both approaches are resource-intensive and time-consuming and can hinder the successful application of CNNs (Leitão et al., 2018; Kattenborn et al., 2019b).

One alternative solution is using crowd-sourced plant photos from citizen science species identification platforms, such as iNaturalist and Pl@ntNet (Boone and Basille, 2019; Di Cecco et al., 2021; Joly et al., 2016; Affouard et al., 2017). These platforms host millions of vascular plant photos annotated with species labels, representing a valuable resource for training computer vision models (Joly et al., 2016; Van Horn et al., 2018). Currently, the iNaturalist platform hosts more than 39 million globally distributed and annotated photographs of vascular plant species. iNaturalist further allows users to identify plant species manually or via an integrated computer vision model. The submitted observations by volunteers are then evaluated and a research-grade label is assigned when two-thirds of the community agree on the species identification. Similarly, Pl@ntNet, which includes over 12 million observations, offers an integrated computer vision model for identifying plant species and a validation process based on community consensus (Joly et al., 2016). Both platforms contribute their validated data to the Global Biodiversity Information Facility (GBIF), an international repository for



open-access biodiversity information (GBIF, 2019).

Understanding where exactly the plant is located in an image is crucial for many applications. Precise localization enables the isolation of the plant from its background, allowing for the extraction of key morphological features such as shape, texture, and pattern. This detailed spatial information is essential for distinguishing between individual plants, even among those of the same species, for assessing plant health, growth patterns, and responses to environmental conditions. In scenarios where plants are densely clustered or overlapping with other objects, accurate localization ensures that analyses focus solely on the plant itself rather than extraneous elements.

This need for spatial precision is particularly pronounced in applications that use UAVs for species identification and ecological monitoring. UAV images often capture complex landscapes where multiple species intermix, making it difficult to discern individual plants using only image-level labels. Image segmentation, which involves partitioning an image into distinct regions corresponding to different objects, provides the necessary pixel-level detail. By isolating individual plants from the background and neighboring species, segmentation enhances the ability to analyze fine-grained characteristics and supports more accurate species differentiation and monitoring.

Despite the abundance of openly available crowd-sourced plant photographs from platforms such as iNaturalist and Pl@ntNet, most of these datasets provide only weak, image-level species labels. While these labels are sufficient for training image classification models to predict whether a species is present in a photograph, they do not provide the detailed, pixel-level annotations (masks) required for training segmentation models. This lack of masks prevents the effective isolation of plant parts and the extraction of fine-grained morphological features, which are crucial for applications like UAV-based species identification. Consequently, these datasets are fundamentally unsuitable for training supervised segmentation models, creating a critical gap between the vast availability of plant photographs and the detailed mask annotations required.

To bridge this gap, we propose a novel workflow for automatically generating high-quality masks from simple plant species labels using the Segment Anything Model (SAM), a state-of-the-art foundation model designed for generic segmentation tasks, in combination with Gradient-weighted Class Activation Mapping (Grad-CAM) (Kirillov et al., 2023; Selvaraju et al., 2017). The SAM model can automatically segment objects in photos (e.g. plant parts of a given species in an iNaturalist image) based on one or a few sample point locations within example photographs. To automate the process of creating sample point locations for a given plant species in a photograph, we employed supervised image classification models and the Grad-CAM technique. Grad-CAM enables the identification of image regions that are important to correctly predict a given class (here a plant species); thus we use sample location of such regions as input for the segmentation mask creation with SAM. This approach facilitates the automated creation of masks for crowd-sourced plant photographs.

In this study, we demonstrate an end-to-end workflow that transforms the simple labels of crowd-sourced plant photos from iNaturalist and Pl@ntNet into segmentations masks,



making them directly useable for training segmentation models for plant species mapping. We then demonstrate how these generated masks can be used to train CNN-based semantic segmentation encoder-decoder models, which can then be applied to UAV orthoimagery for accurate and large-scale segmentation of plant species (Kattenborn et al., 2021b; Bayraktar et al., 2020; Brandt et al., 2020). We further test how this workflow can be used to scale training data collection for broader UAV applications, offering an automated and scalable approach for image mask collection that minimizes the need for extensive manual annotations. To evaluate the effectiveness of the proposed workflow, we tested it on a tree species dataset acquired from the MyDiv experimental site in Bad Lauchstädt, Germany (Ferlian et al., 2018). This site comprises ten temperate deciduous tree species in varying mixtures, making them ideal for testing species identification tasks.

## 2 Methods

### 2.1 Data acquisition and pre-processing

#### 2.1.1 Study site and drone data acquisition

We evaluated the effectiveness of our proposed approach for segmenting tree species at the MyDiv experimental site. The MyDiv experimental site, located at the Bad Lauchstädt Experimental Research Station of the Helmholtz Centre for Environmental Research–UFZ in Bad Lauchstädt, Saxony-Anhalt, Germany (latitude 51°23' N, longitude 11°53' E), comprises 20 monoculture plots of ten tree species (two per species). The species comprise *Acer pseudoplatanus*, *Aesculus hippocastanum*, *Betula pendula*, *Carpinus betulus*, *Fagus sylvatica*, *Fraxinus excelsior*, *Prunus avium*, *Quercus petraea*, *Sorbus aucuparia*, and *Tilia platyphyllos* (Ferlian et al., 2018). Each plot is 11 m × 11 m, with 140 trees planted one meter apart, totaling 2,800 tree individuals in the entire study area (Fig. 1).

We acquired UAV-based RGB aerial images using a DJI Mavic 2 Pro drone using the DroneDeploy flight planning software (version 5.0, USA). The data was acquired in July 2022, representing the peak growing season (Fig. 1). The flight was carried out at an altitude of 16 m, with a 90% forward overlap and a 70% side overlap, achieving a ground sampling distance of approximately 0.22 cm per pixel. We processed orthoimage of the MyDiv experiment using the collected UAV images using Metashape (version 1.7.6, Agisoft LLC).

Using the generated orthoimage, we created reference data of the target species in an earlier study for the MyDiv experiment site that evaluated the accuracy of CNN-based segmentation models for segmenting tree species in UAV orthoimages(Soltani et al., 2023). This reference data was created by manually delineating the canopies of the tree species in the UAV orthoimages using QGIS (version 3.32.3). Given the laborious effort to create such reference masks at high quality, we created diagonal transects for each plot measuring 20 m in length and 2 m in width, instead of annotating the entire plot.



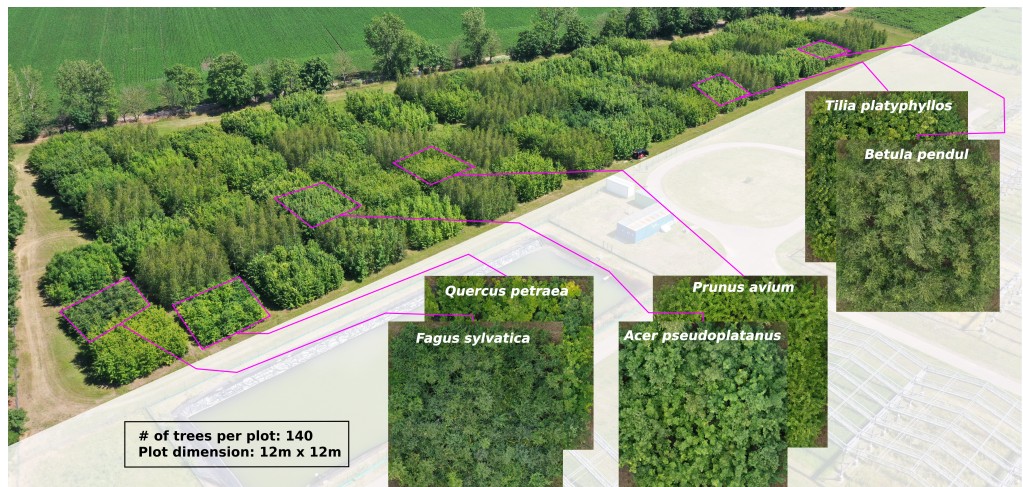

Figure 1: Overview of the MyDiv experimental site, highlighting six monoculture plots in close-up. The site is located at 51.3916° N, 11.8857° E.

### 2.1.2 Citizen science data

To compile the tree species training dataset, we queried citizen science plant observations from the iNaturalist and Pl@ntNet via the Global Biodiversity Information Facility (GBIF) database using the scientific names of the target tree species. We retrieved iNaturalist data using the R package rinat (version 0.1.8), which provides an application programming interface (API) for iNaturalist. We acquired Pl@ntNet data directly from GBIF as tabulated observation records through an R pipeline that downloads the plant photographs using the provided URLs. We restricted the iNaturalist dataset to photographs taken between May and September to avoid photographs in leaf-off conditions, improving their comparability with the UAV orthoimages. Given that plant photographs within the Pl@ntNet dataset are initially focused on the green parts of the plants such as leaves, it was unnecessary to perform any date-based filtering on the Pl@ntNet data.

The number of photographs available for each species varied across our two datasets. We were able to download between 893 and 10,000 photographs per species (mean = 7957) from the iNaturalist platform. Likewise, the Pl@ntNet platform provided between 221 and 3,304 photographs per species (mean = 2,238). Complete details regarding the number of downloaded photos per species are provided in Table A1 in the Appendix.

We preprocessed all photos to ensure a uniform shape by cropping each image to a rectangular shape based on the shorter side and resampling them to 512 × 512 pixels. The RGB values were normalized within the range [0, 1].

## 2.2 Segmentation mask creation

Various ongoing efforts are being made for automated segmentation of objects in images, with Facebook's Segment Anything Model (SAM) being one of the most widely used and efficient



techniques (Kirillov et al., 2023). This model segments objects based on boxes and/or points as inputs. We automated the process of generating input points for the SAM model by utilizing the feature attribution method Grad-CAM (Selvaraju et al., 2017). Grad-CAM attributes a decision of an existing model to the pixels of an input image. The citizen science photographs with their simple species labels allowed us to train image classification models for the target tree species to predict if one of the tree species is somewhere in the photograph. Using Grad-CAM, we located the pixels that were important for the model to reveal the approximate location of the individual within the image. Then, we sampled points from these image regions as input for the segmentation mask generation using SAM.

For training the image classification model, we used the EfficientNet-V2 Large architecture (Tan and Le, 2021). The final classifier layer was adjusted to correspond to the number of tree species plus an additional class for the background. In particular, the default fully connected layer was substituted with a linear layer comprising eleven output units, which map to each distinct plant species or background class. To achieve a balanced dataset for training, we selected 4,000 photographs per class. For those species with fewer available photographs, we duplicated the existing photographs. A data augmentation was applied to all photographs to increase generalization and to minimize the redundancy of duplicated photographs. The data augmentation included random horizontal and vertical flips, color jitter, random cropping, and random erasing with a probability of 20%. The image classification model was implemented using the PyTorch framework and trained on a high-performance GPU system (NVIDIA A6000 with 48GB RAM). We allocated 80% of the labeled reference data to training and remaining 20% for validation. We used AdamW Optimizer (learning rate of 0.001 with a weight decay of 1e-4). To enhance the model's learning process and improve generalization, a dynamic learning rate schedule was applied using a OneCycleLR scheduler, with a maximum learning rate defined at 0.01 (Smith, 2018). We trained the image classification model using a batch size of 16 and 150 epochs.

We used the final model with the lowest validation loss to generate Grad-CAM heatmaps. After several tests, we found that the original Grad-CAM implemenentation revealed meaningful outputs and had high computational efficiency (Selvaraju et al., 2017). Output values ranged from 0 to 1, with higher values indicating greater importance for a given species' identification. To enhance the precision of input point selection, we applied a contour-based sampling method that restricted point placement to regions with an activation probability threshold >0.6. After multiple tests, we found that placing two input points per contour yielded optimal segmentation performance. The sampled points were used as input for SAM, which subsequently automatically generated segmentation masks for the citizen science plant photographs.

## 2.3 Harmonizing citizen science photographs with UAV images

We performed several preprocessing steps to improve the consistency between the perspective of the citizen science photographs and UAV orthoimagery. A significant challenge in using citizen science plant photographs is that they often include understory vegetation and background



elements, such as shrubs, herbs, and forest ground. The latter are typically not visible in UAV imagery, as they predominantly capture the upper canopy structure. These background elements during training can introduce unwanted variance and increase the complexity of the segmentation model, potentially leading to misclassification and reduced performance.

To simulate the top-down canopy perspective for the citizen science photographs, we replaced the backgrounds in the crowd-sourced photographs with background images derived from the UAV orthomosaic. We used the masks derived with Grad-CAM and SAM to automatically substitute the background of the citizen science photos across the entire dataset. This approach preserved the shape and structure of the target species in the foregrounds, while ensuring that the backgrounds matched the visual characteristics captured in the UAV orthoimage. For the background class, we manually extracted a total of 1,879 high-resolution close-up images from the UAV orthoimage, including exposed soil, herbaceous vegetation, and leaf litter ensuring comprehensive coverage of various background types in the study area.

Another preprocessing step involved zoom-outs of the original plant photographs. Citizen science photographs often include close-ups of plants and their leaves. To align such photographs with the often more distant UAV image acquisition geometry, the entire citizen science training dataset was augmented through zoom-out operations. Specifically, we duplicated each photograph and zoomed out the plant foreground by 60%. This approach ensures that our training dataset includes both the original and zoomed-out photographs, better aligning with the varying acquisition geometries.

One common limitation of our automated workflow for mask generation was that it occasionally failed to detect the entire plant within a photograph, instead detecting only small fragments of the foreground plant (e.g., a single branch or leaf). To exclude these incomplete masks and their corresponding photographs, we filtered out all masks when the detected plant in the foreground was less than 30% of the total photograph area. This threshold was empirically determined to strike a balance between retaining meaningful samples and removing erroneous data.

Citizen science photographs exhibit substantial variability in acquisition perspectives and settings compared to UAV imagery. UAV images typically capture tree canopies from a consistent bird's eye view at uniform distances, whereas citizen science photographs often include close-up views of leaves, horizontal shots of trunks, or landscape views. Previous work by Soltani et al. (2022, 2023) demonstrated that excluding photos based on acquisition distance, such as too close to plants or far away showing landscapes, photos mainly showing tree trunks improve the precision of species segmentation in UAV orthoimage. To enhance the quality of training data, we applied filters based on acquisition distance and the presence of tree trunks. Since the metadata for these attributes is not available on citizen science platforms, we developed a CNN-based regression and classification model to predict acquisition distances in meters and detect the presence of the trunk. The models were developed in our previous study on tree species, and we used them in the current study without any additional fine-tuning. For an in-depth explanation of the methodology, refer to (Soltani et al., 2022). Based on these predictions, we excluded photos with predicted acquisition distances of less than 0.2 m





or greater than 20 m, as well as photos for which a probability of including a tree trunk was greater than 0.5. After filtering, 65,024 of the original 112,018 photographs were retained for tree species segmentation. Fig. 2 illustrates examples of processed images and highlights their visual differences compared to UAV orthoimages.





Figure 2: Example citizen science-based photographs derived from iNaturalist and tiles of UAV orthoimages (512 * 512 pixels) for the ten tree species in the MyDiv experiment.



## 2.4 CNN-based plant species segmentation using an encoder-decoder architecture

For our segmentation architecture, we chose U-Net (Ronneberger et al., 2015), which is the most widely applied segmentation architecture in remote sensing image segmentation (Kattenborn et al., 2021b). It is implemented as an encoder-decoder network, where the encoder captures hierarchical feature representations and the decoder reconstructs spatial details to generate a dense prediction map. Skip connections link the corresponding encoder and decoder layers, allowing the model to combine high-level semantic information with fine-grained spatial details. This architecture produces semantic segmentation by predicting a class for each input image pixel.

The implemented U-Net comprises four levels of convolutional blocks, each containing two convolutional layers followed by batch normalization and ReLU activation. The encoder progressively compresses the feature maps and reduces their spatial dimensions using max-pooling operations, while the decoder restores the spatial resolution by applying transposed convolutions. Like the image classification model, the output layer includes eleven units representing ten tree species and one background class. A softmax activation function maps the extracted features to class probabilities. Using a maximum function, the pixels in the final segmentation outputs are assigned to the class with the maximum probability (Fig. A1).

After filtering the training data (see section 2.3), the number of photos per tree species ranged from 2,342 to 13,303 samples: *Acer pseudoplatanus* (6,991), *Aesculus hippocastanum* (7583), *Betula pendula* (6,129), *Carpinus betulus* (7,849), *Fagus sylvatica* (6,873), *Fraxinus excelsior* (9,094), *Prunus avium* (4,883), *Quercus petraea* (6,344), *Sorbus aucuparia* (7,792), and Tilia *platyphyllos* (1,486), and background (1,879). Similar to the image classification model training, to avoid any effects of class imbalance which can lead to bias in the segmentation model, we sampled 4,000 photos per class. We applied sampling with replacement for the classes with fewer photos. To increase the variance of duplicated photos, we applied data augmentation including vertical and horizontal flips, random brightness adjustments with a maximum delta of $\pm 10\%$ (0.1), and contrast alterations within a range of 90% to 110% (0.9 to 1.1) of the original training photographs. We partitioned the training photographs into 80% training and 20% validation sets for the evaluation of the segmentation model.

We trained the U-Net architecture using the RMSprop optimizer with a learning rate of 0.0001 and a custom Focal Tversky loss to handle class imbalance. The Tversky index, with $\alpha = 0.5, \beta = 0.5, and \gamma = 1.5$, emphasized false negative and false positive penalties from the loss computation. A custom Dice coefficient was used for performance evaluation. We trained the models with a batch size of 10 over 80 epochs. The final performance of the segmentation model was assessed using independent reference data derived from visual interpretations of UAV orthoimage transects.



**Raw photos**

**Heatmap**

**Contour + Point**

**Segmentation mask**

**Background replaced**

**Segmentation model for UAV image**

Figure 3: Schematic workflow of the segmentation mask generation. Heat maps are generated using the Grad-CAM approach on the species classification model. The resulting heatmaps are used to extract sample points for the SAM foundation model, which results in precise segmentation masks. These segmentation masks are then used to replace the background with a background corresponding to the UAV imagery. This modified citizen science photos and segmentation masks are then used to create segmentation models which are applied to UAV orthoimages.





## 3 Results

The segmentation model trained using mask data from our new automated labeling approach yielded varying F1 scores for different tree species, utilizing UAV orthoimages (Fig. 4). At the plot level, high model performance (mean F1 >0.6) was observed for *Acer pseudoplatanus* and *Tilia platyphyllos*. This was followed by moderate performance (mean F1 score: 0.35–0.56) for *Aesculus hippocastanum*, *Carpinus betulus*, *Fagus sylvatica*, and *Quercus petraea*. Low segmentation performance (mean F1 score < 0.35) was observed for *Sorbus aucuparia*, *Prunus avium*, *Fraxinus excelsior*, and *Betula pendula*.

We observed large differences in the confusion between species, where some species were rather randomly and sometimes systematically confused with each other. For instance, we observed many false positives for *Prunus avium*, which was in fact *Aesculus hippocastanum* (25.4%), *Betula pendula* (33.4%), *Carpinus betulus* (28.7%), *Fagus sylvatica* (31.6%), and *Fraxinus excelsior* (21.7%) (Fig. 4).

Furthermore, *Sorbus aucuparia* was often confused for *Fraxinus excelsior*, with the former incorrectly classified at a confusion rate of 42.6% (Fig. 4). Both the model performance and the confusion were related to leaf size, where the performance of the model declined as the leaf size decreased in the tree species dataset.



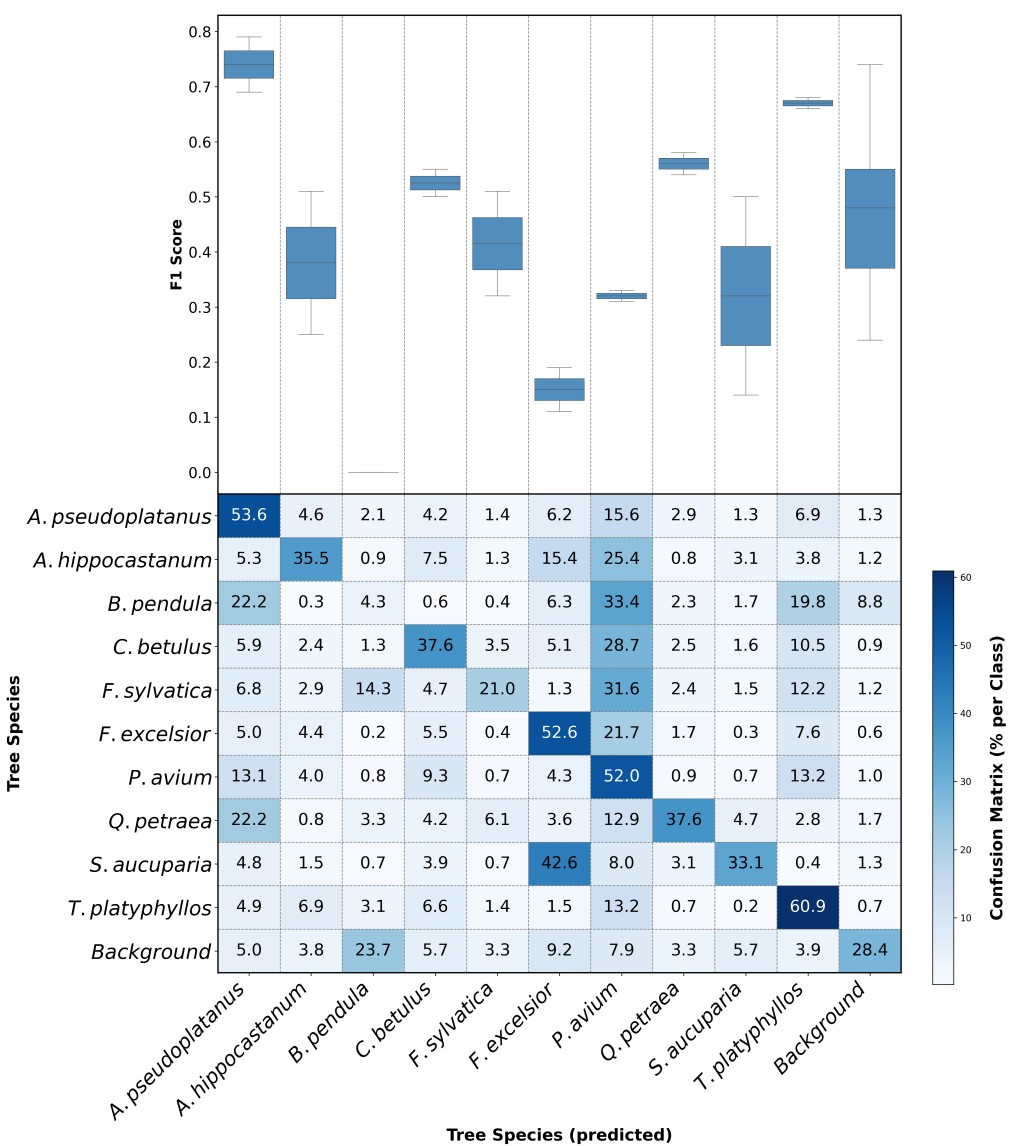

Figure 4: F1-scores and confusion matrix for segmenting tree species in UAV orthoimagery.

## 4  Discussion

### 4.1  Automated mask generation of crowd-sourced plant photos using the SAM foundation model and Grad-CAM technique

Labeling training data for computer vision, particularly for vegetation monitoring using remote sensing data, remains a persistent challenge, primarily due to the complexity and scale of datasets required for such applications (Singh and Surasinghe, 2024; Gillespie et al., 2024; Bah and Hafiane, 2018). This challenge is more pronounced for segmentation tasks that re-





quire mask labels (Maß and Alirezazadeh, 2025; Illarionova et al., 2022; Kattenborn et al., 2021a). Our proposed method attempts to address this issue by automating the mask generation process through the integration of the SAM foundation model with the Grad-CAM technique.

In our study, we focussed on using two Grad-CAM-derived input points as the basis for mask generation. Although this approach proved effective, alternative input strategies, such as polygonal annotations or directly using the most activated regions heatmaps, present opportunities for further exploration. Each method has unique trade-offs in terms of computational complexity, mask quality, and suitability for different applications. For example, polygonal annotations might better capture complex shapes, while thresholded heatmaps could provide more generalized masks for species with diffuse features. Exploring these strategies could enhance the adaptability of our methodology to a wider range of plant species and photographic conditions.

One of our notable findings was the delineation precision of the automated workflow for mask generation. From the Grad-CAM-generated input points, SAM was able to follow leaf contours with high precision. Achieving this level of delineation precision through manual efforts is infeasible, especially when handling large datasets. Therefore this automated mask generation addresses one of the most labor-intensive steps in ecological monitoring (Kattenborn et al., 2021b; Katal et al., 2022; Maß and Alirezazadeh, 2025). The capability to automatically and efficiently generate detailed masks removes barriers to annotating large datasets, paving the way to leverage large, heterogeneous datasets for remote sensing applications.

The mask generation approach presented further enabled to tailor the training data to the remote sensing scene. Here, we modified the background of the training data with the background that was observed in UAV imagery. This was only possible due to the precise segmentation masks created using the presented workflow. Previous modeling attempts (results not shown) showed that the presence of various image components, such as understory, may complicate a model transfer to the UAV scale. Instead, removing the original background with the UAV-based background greatly improved the model performance. This innovation highlights that datasets with extreme variance can be tailored to specific remote sensing applications.

By automating segmentation mask generation, our proposed methodology also significantly improves the utility of citizen science data for ecological research. Our approach enables the exploitation of the huge and rapidly growing availability of citizen science plant photographs for remote sensing-based vegetation monitoring. Citizen science platforms such as iNaturalist and Pl@ntNet already include plant photographs of more than 170,000 and 75,000 plant species, respectively, with many species already being observed hundreds to thousands of times (ina, 2025; pla, 2025). Automatically labeled citizen science photos as presented in this study can be used as standalone datasets or augment existing training data. Citizen science data further comes with a high variability in image geometries, acquisition dates, and plant status, making it a rich and diverse source of training data. Lastly, both





citizen science data collection and UAV data acquisition are effective across diverse land-scapes (Soltani et al., 2023; Katal et al., 2022; Schiller et al., 2021) and scalable across wide geographies (Gillespie et al., 2024), raising the exciting possibility of real-time, landscape-level vegetation monitoring. This further offers the public unique opportunities to support biodiversity monitoring and conservation by transforming their contributed data into actionable insights for large-scale environmental applications.

Beyond vegetation monitoring, the presented workflow of an automated segmentation mask generation holds transformative potential for a variety of supervised computer vision tasks across multiple domains. Automating the mask generation process not only accelerates the annotation workflow but also significantly improves the scalability and robustness of deep learning models, enabling them to be trained across large datasets where manual annotation would be infeasible.

## 4.2 Filtering of citizen science data for UAV-related applications

The filtering of citizen science photographs was crucial for aligning ground-based observa-tions with UAV imagery. This process involved selecting photos captured during the growing season (May to September), ensuring appropriate acquisition distances, and excluding im-ages dominated by tree trunks. Since deciduous tree photos taken during winter primarily show leafless branches, filtering for seasonal relevance ensured consistency with UAV canopy imagery captured in full foliage.

A key limitation of automated mask generation was the partial detection of plant features (e.g., isolated leaves or stems). To address this, we filtered out photographs with less than 30% background coverage, removing incomplete detections and preventing the dataset from being skewed toward the background class. This balancing step reduced the dominance of background pixels, improving label accuracy and ensuring that training data more closely represented features observable in UAV orthoimages. These filtering measures enhanced data quality, minimized irrelevant inputs, and improved overall model performance.

## 4.3 Segmentation performance

Segmenting temperate broadleaf tree species using UAV imagery presents a complex challenge due to leaf form similarities between many species (Fig. 2). This complexity was evident in the confusion patterns observed in this study, particularly among species such as *Sorbus aucu-paria* and *Fraxinus excelsior*. Despite having different leaf arrangements, these species appear remarkably similar at the current resolution of UAV imagery (0.22 cm) in the absence of flow-ers and fruits for *Sorbus aucuparia* (Fig.2). This similarity in leaf shape makes them difficult to visually distinguish, even in centimetre-scale orthomosaics. Although *Fraxinus excelsior* possesses pointy leaves, as visible in citizen science photographs, these subtle distinctions are barely visible in UAV orthoimages, further compounding the classification challenges. These findings highlight that some species require even higher-resolution UAV data to capture the subtle morphological details critical for species differentiation.



The resolution of UAV imagery is critical for accurate segmentation, as it allows for the detection of fine morphological features such as leaf shape and branching patterns. In this study, species with higher model performance, such as *Acer pseudoplatanus* and *Tilia platyphyllos*, benefited from their large, distinct leaves, which were more readily detectable in UAV orthomosaics. However, the presence of other species of the respective genus with a similar leaf shape, e.g. *Acer planatonides* or *Tilia cordata*, that were not present in MyDiv but may co-occur in real-world forests, may lead to the similar confusion as between *Fraxinus excelsior* and *Sorbus aucuparia*. Still, in the present study, the high F1 scores for *Acer pseudoplatanus* (mean F1: 0.74) and *Tilia platyphyllos* (mean F1: 0.67) underscore the advantage of pronounced morphological patterns. However, the model's performance declined significantly for species with smaller or morphologically similar leaves, such as *Betula pendula* and *Fraxinus excelsior*. These species demonstrated very low mean F1 scores, indicating a notable limitation in resolving fine-scale distinctions. Higher-resolution UAV images finer than 0.22 cm could allow the model to capture species-specific details that are currently apparent, and subtle differences in leaf shape and arrangement. These findings align with a previous study using the MyDiv dataset, where small canopy areas decreased segmentation performance(Soltani et al., 2023).

Further model improvements might be possible with increasing availability of citizen science photographs. Here, for most species we could obtain between 2,342 to 13,303 photographs per species. The increasing size of citizen science data will provide more diverse representations of species across different habitats and seasons (Boone and Basille, 2019). Together with filtering such data and further tailoring it to the UAV perspective, incorporating additional and high-quality citizen science photographs could address current limitations related to the underrepresentation of specific species and enhance the model's ability to generalize to new environments.

In addition to data improvements, leveraging advanced segmentation models could address some of the limitations observed in this study. While U-Net has been effective in segmentation tasks, here, its performance is constrained in scenarios with similar morphological features and complex canopies. More complex architectures or methods, such as transformer or deeper CNN architectures, which integrate multi-scale feature extraction and attention mechanisms, offer promising alternatives (Li et al., 2024). We assume that particularly in concert with higher resolution data, such methods could significantly enhance segmentation accuracy for challenging species like *Betula pendula* and *Fraxinus excelsior*.

## 5 Conclusion

This study demonstrates the value of citizen science photographs for remote sensing-based plant species identification. We showed that the simple species annotations of citizen science projects can be automatically used to create segmentation masks with high precision. These segmentation masks can then be used to train segmentation models that allow for the localization of plant species in UAV imagery. Despite the inherent complexity of segmenting



broadleaf tree species, the model achieved an overall acceptable performance. This performance is enhanced by filtering the citizen science data. Here, we filtered photographs that are not similar to the UAV perspective, including photographs that are too close or too far from the plant or photographs that focus on stems of trees. We also demonstrated that the citizen science photographs can be further tailored to the remote sensing imagery by replacing the background of the photograph using the automatically generated segmentation masks with a typical background of the UAV images. By bridging citizen science with advanced remote sensing and machine learning, this study lays a foundation for inclusive and scalable biodiversity assessments, supporting efforts to understand and preserve Earth's ecosystems.

# 6  Data and code availability

The code used in this study is publicly accessible via our GitHub repository at https://github.com/salimsoltani28/Flora_Mask. The data supporting the findings of this research is available on Zonodo at https://zenodo.org/uploads/10019552 (Kattenborn and Soltani, 2023).

# 7  Author contributions

SS: conceptualization, methodology, formal analysis, data curation, visualization, and writing – original draft preparation. LEG: mentoring, methodology, writing – review and editing. MEA: provided lab resources, supervision, hosting SS as research visit, writing - review and editing. OF: provided resources and contributed to writing – review and editing. NE: provided resources and contributed to writing – review and editing. HF contributed to funding acquisition, supervision, and writing – review and editing. TK contributed to conceptualization, methodology, data collection, funding acquisition, data curation, resource acquisition, supervision, and writing – original draft preparation.

# 8  Declaration of competing interest

The authors declare that they have no known competing financial interests or personal relationships that could have appeared to influence the work reported in this paper.

# 9  Acknowledgements

SS and TK acknowledge funding by the German Research Foundation (DFG) under the project BigPlantSens (Assessing the Synergies of Big Data and Deep Learning for the Remote Sensing of Plant Species; Project number 444524904) and PANOPS (Revealing Earth´s plant functional diversity with citizen science; project number 504978936). SS acknowledges financial support from the XR Future Forests Lab, Faculty of Environment and Natural Resources, University of Freiburg (with corresponding funding from the Eva Mayr-Stihl Stiftung). SS



also acknowledges funding for a research visit from the Young Biodiversity Graduate School at the German Centre for Integrative Biodiversity Research. HF acknowledge financial support by the Federal Ministry of Education and Research of Germany (BMBF) and by the Saechsische Staatsministerium für Wissenschaft, Kultur und Tourismus in the program Center of Excellence for AI-research "Center for Scalable Data Analytics and Artificial Intelligence Dresden/Leipzig", project identification number: ScaDS.AI. NE and OF acknowledge funding by the Deutsche Forschungsgemeinschaft DFG (German Centre for Integrative Biodiversity Research, FZT118; and Gottfried Wilhelm Leibniz Prize, Ei 862/29-1). LEG acknowledges funding by the NSF Graduate Research Fellowship DGE-1656518 and the TomKat Graduate Fellowship for Translational Research. Lastly, ME-A acknowledges funding by the Office of the Director of the National Institutes of Health's Early Investigator Award (1DP5OD029506-01), the U.S. Department of Energy, Office of Biological and Environmental Research (DE-SC0021286), and by the U.S. National Science Foundation's DBI Biology Integration Institute WALII (Water and Life Interface Institute, 2213983).

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





## A Appendix

### A1.1 Segmentation model archeticture

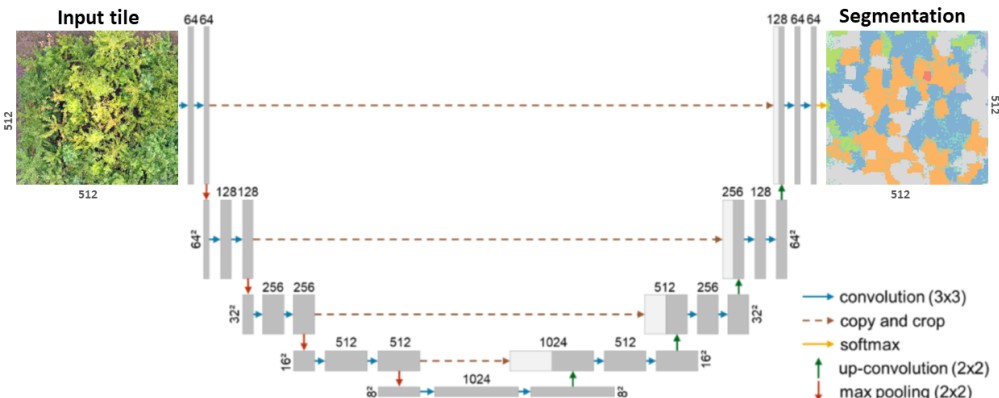

Figure A1: U-Net architecture adopted from Soltani et al. (2023) for segmenting tree species in UAV orthoimages (Ronneberger et al. (2015))

### A1.2 Citizen science data availability

Table A1: Number of downloaded photographs for selected tree species from the iNaturalist and Pl@ntNet datasets.

| No. | Species | iNaturalist | Pl@ntNet | Note |
|-----|---------|-------------|----------|------|
| 1 | Acer pseudoplatanus | 10000 | 3205 | |
| 2 | Aesculus hippocastanum | 9997 | 1444 | |
| 3 | Betula pendula | 10000 | 1308 | |
| 4 | Carpinus betulus | 9999 | 2633 | |
| 5 | Fagus sylvatica | 9999 | 3304 | |
| 6 | Fraxinus excelsior | 10000 | 3130 | |
| 7 | Prunus avium | 6265 | 3022 | |
| 8 | Quercus petraea | 2419 | 221 | |
| | Quercus robur | 9993 | - | Supplement species |
| 9 | Sorbus aucuparia | 10000 | 2730 | |
| 10 | Tilia platyphyllos | 893 | 1449 | |