# Peer review of "From Ground Photos to Aerial Insights: Automating Citizen Science Labeling for Tree Species Segmentation in UAV Images"

_EGUsphere, 2025_

## Author Comment (AC1)

| Comments Reviewer #1 | | | |
|---|---|---|---|
| **ID** | **Line** | **Comment** | **Response** |
| 1 | 184-188 | Other than learning rate, batch size, and epoch, did you tune other parameters? Also, for learning rate, batch size, and epoch, it is better to test with a wider range of values to evaluate model performance before narrowing them down to a specific range. Also, for model training, did you use k-fold cross-validation for hyperparameter tuning? If so, what is the k-fold value did you use? This needs to be clarified. | We thank the reviewer for pointing this out. Yes, we tested different hyperparameter settings both in this study and in our earlier work (Soltani et al., 2024). The reported parameter settings yielded improved results and were therefore adopted. In the revised version, we will describe the hyperparameters and how we selected them in more depth.
We did not use k-fold cross-validation. The models were evaluated on an entirely independent test dataset (see Section 2.1.1). |
| 2 | 239-243 | The prediction of acquisition distance seems skeptical. In citizen science data, people use various cameras and may set various zooming modes when capturing photos, it is hard to predict acquisition distance just from the photo itself; thus, distance thresholds of 0.2 m and 20 m seem skeptical. In the earlier paragraph, authors use an area threshold of 30% to filter out some photos. Should a similar method be used to filter out photos with large amounts of tree trunk/branch? | We understand the reviewer's skepticism about estimating camera-to-plant distance from a single photo. Inferring absolute distance is indeed challenging without known camera parameters. Our approach was intended to exclude extremely close-up photos showing individual leaves, or very distant photos showing broad landscapes. It does not aim to provide precise distance estimations but rather to filter out these two extreme cases. The applied threshold effectively removed such images, allowing us to include photos taken at distances commonly found in close-range UAV imagery. We will include visual examples in the supplementary information to transparently demonstrate the effectiveness of this method, which was already successfully applied in Soltani et al. (2019). |
| 3 | 278-284 | Did you use k-fold cross-validation to train the model? If so, the k-fold value you used should be reported. | We did not use k-fold cross-validation during model training. Final model evaluation was performed using manually delineated reference data from UAV images that were |

| | | | completely excluded from the training process (see Section 2.1.1). We will make this clearer in the revised manuscript. |
|---|---|---|---|
| 4 | 286-301 | The classification performance seems to be low for various species. Citizen science data helps reduce time and labor in reference data collection; however, we also need to make sure output data are accurate and usable. With this low accuracy, what do authors suggest for future works? Should we incorporate some UAV-based high accuracy labelled data in the model together with citizen science data to improve classification accuracy? Also, the hyperparameter tuning seems not to be well-performed in your deep learning model training, I recommend conducting a more exhaustive tuning and trying different deep learning architecture to see if the classification results are improved | We acknowledge the reviewer's concerns regarding the partially moderate segmentation accuracy and appreciate the forward-looking suggestions. First of all, we would like to highlight that using citizen science data for drone-based remote sensing is still in its infancy, and we are just pioneering the possibilities. This study is not about providing an operational technology, but rather about exploring methodological ways to harness citizen science data and its potential for drone-based mapping.

 Here, we demonstrate this potential in a very complex scenario, where several broadleaved tree species with very similar leaf forms are present. Given this pioneering character and the complexity of the case study, we are of the opinion that the results are groundbreaking and open up possibilities for a series of follow-up studies. Clearly, there are many aspects that can be improved and explored in greater depth (see also our Outlook section in the Discussion). In the revised manuscript, we will make it clearer that this study is of a pioneering nature and focuses on method development rather than providing a ready-to-use solution.

 We explored several strategies to improve segmentation accuracy across all tree species, including data augmentation, modifications of photograph backgrounds and scaling, hyperparameter tuning, and adjustments to model architectures. However, visual similarities among certain species led to trade-offs, improving accuracy for one species sometimes decreased it for others. |

| | | | Over several months, we conducted a thorough model ablation study, and the results presented here are the final outcome. In the revised manuscript, we will provide more information on these model ablations. |
|---|---|---|---|
| 5 | | One of the main reasons that cause low segmentation accuracy in this study could be the difference in the spatial resolutions between citizen science photos and UAV images. One possible solution for this discrepancy could be that during your segmentation model training, authors may want to manipulate/resample citizen science photos to different resolutions, including the 0.22 cm resolution of the UAV image, and incorporate features extracted from these layers into the final segmentation prediction to help improve the final segmentation results (see below paper with similar idea, note: this is not a reviewer's paper). Martins et al., 2020. Exploring multiscale object-based convolutional neural network (multi-OCNN) for remote sensing image classification at high spatial resolution. https://doi.org/10.1016/j.isprsjprs.2020.08.004 | We would first like to point out that for several species in the study, we have very high segmentation accuracy and overall model performance (e.g., F1 score above 0.5 for *Acer pseudoplatanus*, *Tilia platyphyllos*, *Quercus petraea,* and *Carpinus betulus*). This is particularly striking given that we did not acquire any specific training data and that the approach is entirely based on crowd-sourcing. It is important to note that this study is primarily about providing a methodological framework and showcasing the potential of such an approach. It is expected that this method will not (yet) work for all species, as this is truly pioneering work. In fact, we also aim to highlight the limitations of the approach, for instance, where it does not work well, such as with species that have very similar leaves (see Discussion, lines 377–404). Accordingly, in the revised manuscript, we will describe the overall objectives and limitations of this study more clearly. We agree that differences in spatial resolution and perspective present a challenge for our transfer learning approach. In our current implementation, we addressed this in part by downscaling, duplicating, and zooming out the citizen science photos before using them for training (see Section 2.3). This increased the likelihood that the appearance of plant |

| | | | features in citizen science imagery and drone-based imagery would align. |
| | | | Achieving a perfect resolution match is difficult due to variability in ground-level photo distances, image quality, and variation in the drone-based imagery (e.g., due to differences in canopy height). Instead, we applied a generic scaling strategy to reduce the level of detail across all ground photographs. This, combined with data augmentation, helped the model learn more scale-invariant features, which in turn improved generalization to UAV-scale imagery. |
| | | | Thank you for providing the reference on multiscale architectures. From our experience across a range of projects, we found that standard ("vanilla") architectures can learn multiscale phenomena on the fly when sufficient variability is present (see above), and when the model is deep enough. However, testing multiscale models in more depth is certainly promising, particularly if depth information is available, and we will include a discussion of this in the Outlook section of the revised version. |

---

## Author Comment (AC2)

| Comments Reviewer #2 | | | |
|---|---|---|---|
| ID | Line | Comment | Response |
| 1 | | I would recommend the authors to add a workflow chart to help readers understand the various types of methods and data used for the study. There are several AI/ML models employed for various different data processing, including both photographs and UAV imagery. I found it hard to connect the different processing steps, and how different data streams and AI/ML methods are used. | Thank you for this feedback. A flowchart will certainly help guide the reader through the methodological approach. We will therefore add a workflow chart in the revised version. |
| 2 | | Second, not much information is presented in the Results, barely enough to understand the performance of the model. The authors did quite significant work on processing and segmenting the photographs from iNaturealist and Pl@ntNet. However, results about these processing and segmentation are completely missed in the Results. I am nervous the presentation of Results is disconnected with the Methods. Recommend the authors to carefully tie them together, especially, how F1 score, confusion matrix was calculated. The authors mentioned independent transect validation data were identified from UAV imagery, but did not mention where and how those were produced, distribution across species and space etc. I think it is also useful to present the species maps across the experiment plots. | Thank you for this very important feedback! We acknowledge that the original Results section was too brief and appreciate the reviewer's suggestions. We will revise the section to include the missing details and ensure stronger alignment with the Methods section. In particular, we will elaborate on the description of the independent test data creation (transects).

We will include an explanation of how the F1 score and confusion matrix were calculated. Additionally, we will create maps displaying the predictions versus the transect reference data. Together with the flowchart (suggested by the reviewer; see comment no. 1), we assume  that the results and workflow will become more comprehensive. |
| 3 | | Lastly an overall thought, a core advance of using UAV imagery is to provide landscape-scale observations. The authors argued that ultra-high (finer than 0.22 cm) | We agree that canopy structure and form could carry interesting information for species recognition. However, its value is very limited. For instance, in Schiefer et al. (2020), we |

| | | | |
|---|---|---|---|
| | | might be necessary to better segment species from UAV imagery. This statement appears to "false", and ignored that canopy structure and form are important information for species identification, which are not considered in this study. On the other hand, it is cool to generate the initial masks for UAV species identification using photographs, but it might be more useful to iterate over the species segmentation at UAV level, leveraging other information like canopy form and structure, to enlarge training samples at UAV level, instead of forcing UAV data to the same resolution as ground photographs? | found a clear trend that higher spatial resolution imagery (leaf features) clearly outperforms coarser imagery (only including canopy structure) ([https://doi.org/10.1016/j.isprsjprs.2020.10.015](https://doi.org/10.1016/j.isprsjprs.2020.10.015) ). This was also confirmed by various studies (see our review in Kattenborn et al., 2021, [https://doi.org/10.1016/j.isprsjprs.2020.12.010](https://doi.org/10.1016/j.isprsjprs.2020.12.010) ).

 Thus, it can be assumed that detailed features, such as leaf forms, are the cardinal features to differentiate most species. This aligns well with species-recognition protocols used by humans.

 However, note that we do not specifically ignore canopy structure. Many citizen science observations do show entire plants and their canopies. This information is, in fact, included in the model. It seems that we did not clearly visualize and communicate the variability in the underlying citizen science imagery.

 Therefore, in the revised manuscript, we will include a figure in the supplementary information with several examples per species, showing the large variability in the images, including both leaf-level and whole-plant-level photographs. From this, it should become evident that the model could integrate features of the canopy structure if important. Thank you for highlighting this shortcoming! |
| 4 | | I wonder what features the authors used for segmentation? It is clear that the authors used only RGB imagery, but are other indices or transformations incorporated in the SAM segmentation? | The reviewer is correct that we used only RGB image data for both the SAM-based segmentation and the CNN model. We did not compute or incorporate any additional spectral or textural indices, as SAM is primarily trained on RGB imagery and hence, any sophisticated feature engineering does not appear to be necessary. This is confirmed by the overall high quality |

| | | | of the output (see Fig. 3). In the revised manuscript, we will make it clear that we used only RGB imagery as input. |
|---|---|---|---|
| 5 | | The author mentioned that photos/masks from citizen science were 'zoomed' out when applied as training for UAV imagery. What's the resolution after that? Is it comparable to UAV resolution? | Since the original citizen science images had varying resolutions, applying fixed factors for zooming-out did not result in a uniform output resolution. However, our aim was to approximate the UAV resolution as closely as possible. We tested several factors for the zoom-out, and the selected value provided the best model performance . We thank the reviewer for this observation and will include a description of the resolution of the zoomed-out photos and masks in the revised manuscript. |

---

## Author Response (AR1)

Salim Soltani, Chair of Sensor-based Geoinformatics (geosense), Faculty of Environment and Natural Resources, University of Freiburg, salim.soltani@geosense.uni-freiburg.de

To the Executive Editor and Reviewers of Biogeosciences

20.06.25

Ref. No.: egusphere-2025-662- "Automated mask generation in citizen science smartphone photos and their value for mapping plant species in drone imagery "

Dear Dr. Feldman and Reviewers,

Thank you for the constructive comments and the time dedicated to reviewing our manuscript. Your comments helped us improve the quality of the manuscript. We also thank Dr. Feldman for his editorial comments, which we have addressed in the updated manuscript.

We have revised the manuscript accordingly and hope the updated version addresses the shortcomings of the previous version. We look forward to your assessment and the next editorial decision.

Sincerely,

Salim Soltani

(on behalf of the Co-authors, Lauren E. Gillespie, Moises Exposito-Alonso, Olga Ferlian, Nico Eisenhauer, Hannes Feilhauer, and Teja Kattenborn)

|    |         | Reviewer #:                                                                                                                                                                                                                                                                                                                                                                                                                       | 1                                                                                                                                                                                                                                                                                                                                                                                                                                                                                                                                                                                                                                                                                                                                                                                                                                                                                                                                                                                                                                                                                                                                                                                                                                                                                                                                                                                                                                                                                                                                                                                                                                |
|----|---------|-----------------------------------------------------------------------------------------------------------------------------------------------------------------------------------------------------------------------------------------------------------------------------------------------------------------------------------------------------------------------------------------------------------------------------------|----------------------------------------------------------------------------------------------------------------------------------------------------------------------------------------------------------------------------------------------------------------------------------------------------------------------------------------------------------------------------------------------------------------------------------------------------------------------------------------------------------------------------------------------------------------------------------------------------------------------------------------------------------------------------------------------------------------------------------------------------------------------------------------------------------------------------------------------------------------------------------------------------------------------------------------------------------------------------------------------------------------------------------------------------------------------------------------------------------------------------------------------------------------------------------------------------------------------------------------------------------------------------------------------------------------------------------------------------------------------------------------------------------------------------------------------------------------------------------------------------------------------------------------------------------------------------------------------------------------------------------|
| ID | Line    | Comment                                                                                                                                                                                                                                                                                                                                                                                                                           | Response                                                                                                                                                                                                                                                                                                                                                                                                                                                                                                                                                                                                                                                                                                                                                                                                                                                                                                                                                                                                                                                                                                                                                                                                                                                                                                                                                                                                                                                                                                                                                                                                                         |
| 1  | 184-188 | Other than learning rate, batch size, and epoch, did you tune other parameters? Also, for learning rate, batch size, and epoch, it is better to test with a wider range of values to evaluate model performance before narrowing them down to a specific range. Also, for model training, did you use k-fold cross-validation for hyperparameter tuning? If so, what is the k-fold value did you use? This needs to be clarified. | We thank the reviewer for pointing this out. Yes, we tested different hyperparameter settings both in this study and in our earlier work (Soltani et al., 2024), and the parameter settings we originally reported refer to these ideal hyperparameters. In the revised manuscript we describe these hyperparameters and their selection in more detail (Lines 252-267):  "We explored a range of hyperparameters. Specifically, we tested learning rates from 0.00001 to 0.1 and batch sizes between 5 and 20.  Additionally, we evaluated various optimizers (Adam, SGD, AdamW), momentum parameters for SGD (0.4 to 0.99), weight decay for regularization (1e-2 to 1e-5), dropout rates (0.1 to 0.5), and multiple dense layer configurations. We also compared different loss functions, including Cross-Entropy Loss and Focal Loss. Initial experiments showed that the AdamW optimizer with moderate weight decay (1e-4) and no dropout, combined with the dynamic OneCycleLR learning rate scheduler, with a maximum learning rate of 0.01 (Smith, 2018), consistently yielded the most stable and superior convergence.  The optimal hyperparameters identified were a learning rate of 0.001 and a batch size of 16. The final model implementation utilized the PyTorch framework and was trained on a high-performance GPU system (NVIDIA A6000 with 48GB RAM). We partitioned the reference dataset into training (80%) and validation sets (20%)."  We did not use k-fold cross-validation as the models were instead evaluated on an entirely independent test dataset (see Section 2.1.1), a common approach in |

| photos, it is hard to predict acquisition distance just from the photo itself; thus, distance thresholds of 0.2 m and 20 m seem skeptical. In the earlier paragraph, authors use an area threshold of 30% to filter out some photos. Should a similar method be used to filter out photos with large amounts of tree trunk/branch?  Should a similar method be used to filter out photos with large amounts of tree trunk/branch?  Should a similar method be used to filter out photos with large amounts of tree trunk/branch?  Figure A2.  Figure A2.  Lose-up photos showing individual leaves or very distant photos showing broad landscapes. It does not aim to provide precise distance estimations but rather to filter out these two extreme cases. We saw that the applied threshold effectively removed such images while preserving photos taken at distances commonly found in close-range UAV imagery, which can be seen in the series of randomly-selected example citizen science photographs and their predicted distance provided in supplementary figure A2.                                                                                                                                                                                                                                                                                                                                                                                                                                                                                                                                                                                                                                                                                                                                                                                                                                                                                |   |                                                                                                                                                                                                                                                                                                                                                                                                                                             | machine learning (Van Horn et al 2021                                                                                                                                                                                                                                                                                                                                                                                                                                                                                                                                                                                                                                                                                                                                                                                                                                              |
|-----------------------------------------------------------------------------------------------------------------------------------------------------------------------------------------------------------------------------------------------------------------------------------------------------------------------------------------------------------------------------------------------------------------------------------------------------------------------------------------------------------------------------------------------------------------------------------------------------------------------------------------------------------------------------------------------------------------------------------------------------------------------------------------------------------------------------------------------------------------------------------------------------------------------------------------------------------------------------------------------------------------------------------------------------------------------------------------------------------------------------------------------------------------------------------------------------------------------------------------------------------------------------------------------------------------------------------------------------------------------------------------------------------------------------------------------------------------------------------------------------------------------------------------------------------------------------------------------------------------------------------------------------------------------------------------------------------------------------------------------------------------------------------------------------------------------------------------------------------------------------------------------------------------------------------------------------------------|---|---------------------------------------------------------------------------------------------------------------------------------------------------------------------------------------------------------------------------------------------------------------------------------------------------------------------------------------------------------------------------------------------------------------------------------------------|------------------------------------------------------------------------------------------------------------------------------------------------------------------------------------------------------------------------------------------------------------------------------------------------------------------------------------------------------------------------------------------------------------------------------------------------------------------------------------------------------------------------------------------------------------------------------------------------------------------------------------------------------------------------------------------------------------------------------------------------------------------------------------------------------------------------------------------------------------------------------------|
| The prediction of acquisition distance seems skeptical. In citizen science data, people use various cameras and may set various zooming modes when capturing photos, it is hard to predict acquisition distance just from the photo itself; thus, distance thresholds of 0.2 m and 20 m seem skeptical. In the earlier paragraph, authors use an area threshold of 30% to filter out some photos. Should a similar method be used to filter out photos with large amounts of tree trunk/branch?  The prediction of acquisition distance is indeed challenging without known camera parameters. Our approach, which was already evaluated in Soltani et al. (2022, 2024), was intended to exclude extremely close-up photos showing individual leave: or very distant photos showing broad landscapes. It does not aim to provide precise distance estimations but rather to filter out photos with large amounts of tree trunk/branch?  Should a similar method be used to filter out photos with large amounts of tree trunk/branch?  The prediction of acquisition distance is indeed challenging without known camera parameters. Our approach, which was already evaluated in Soltani et al. (2022, 2024), was intended to exclude extremely close-up photos showing individual leave: or very distant photos showing broad landscapes. It does not aim to provide filter out these two extreme cases. We saw that the applied threshold effectively removed such images while preserving photos taken at distances commonly found in close-range UAV imagery, which can be seen in the series of randomly-selected example citizen science photographs under the provided in supplementary figure A2. |   |                                                                                                                                                                                                                                                                                                                                                                                                                                             |                                                                                                                                                                                                                                                                                                                                                                                                                                                                                                                                                                                                                                                                                                                                                                                                                                                                                    |
| distance seems skeptical. In citizen science data, people use various cameras and may set various zooming modes when capturing photos, it is hard to predict acquisition distance just from the photo itself; thus, distance thresholds of 0.2 m and 20 m seem skeptical. In the earlier paragraph, authors use an area threshold of 30% to filter out some photos. Should a similar method be used to filter out photos with large amounts of tree trunk/branch?  Should a similar method be used to filter out photos with large amounts of tree trunk/branch?  Figur A. Employ of disea store, phonyable librarilis, to polled common provided in supplementary figure A2.                                                                                                                                                                                                                                                                                                                                                                                                                                                                                                                                                                                                                                                                                                                                                                                                                                                                                                                                                                                                                                                                                                                                                                                                                                                                                   |   |                                                                                                                                                                                                                                                                                                                                                                                                                                             | Beery et. al. 2022).                                                                                                                                                                                                                                                                                                                                                                                                                                                                                                                                                                                                                                                                                                                                                                                                                                                               |
| "Estimating acquisition distance from photographs using a CNN-based regression model was first introduced in                                                                                                                                                                                                                                                                                                                                                                                                                                                                                                                                                                                                                                                                                                                                                                                                                                                                                                                                                                                                                                                                                                                                                                                                                                                                                                                                                                                                                                                                                                                                                                                                                                                                                                                                                                                                                                                    | 2 | distance seems skeptical. In citizen science data, people use various cameras and may set various zooming modes when capturing photos, it is hard to predict acquisition distance just from the photo itself; thus, distance thresholds of 0.2 m and 20 m seem skeptical. In the earlier paragraph, authors use an area threshold of 30% to filter out some photos. Should a similar method be used to filter out photos with large amounts | Inferring absolute distance is indeed challenging without known camera parameters. Our approach, which was already evaluated in Soltani et al. (2022, 2024), was intended to exclude extremely close-up photos showing individual leaves or very distant photos showing broad landscapes. It does not aim to provide precise distance estimations but rather to filter out these two extreme cases. We saw that the applied threshold effectively removed such images while preserving photos taken at distances commonly found in close-range UAV imagery, which can be seen in the series of randomly-selected example citizen science photographs and their predicted distance provided in supplementary figure A2.  Figure A2: Examples of citizen science photographs (line application distance) in the series of the provided distance provided in supplementary figure A2. |

|   |             |                                                                                                                                                                                                                                          | 11 11 1                                                                                                                                                                                                                                                                                                                                                   |
|---|-------------|------------------------------------------------------------------------------------------------------------------------------------------------------------------------------------------------------------------------------------------|-----------------------------------------------------------------------------------------------------------------------------------------------------------------------------------------------------------------------------------------------------------------------------------------------------------------------------------------------------------|
|   |             |                                                                                                                                                                                                                                          | model achieved an R2 = 0.7 on independent test data. This accuracy indicates reliable performance in predicting acquisition distances from crowd-sourced photographs. An example of the model's predictions and the resulting distance-based filtering is provided in the appendix (Fig. A2)"                                                             |
|   |             |                                                                                                                                                                                                                                          | Concerning tree trunk filtering, we addressed this issue by applying a separately trained classification model, which effectively filtered out photos dominated by bark or woody parts, making additional filtering unnecessary. We made this clear by making it more clear (Lines 327-329):                                                              |
|   |             |                                                                                                                                                                                                                                          | "we developed a CNN-based regression
model to predict acquisition distances in
meters and a separate CNN-based
classification model to detect the presence
of the trunks."                                                                                                                                                                    |
| 3 | 278-
284 | Did you use k-fold cross-validation
to train the model? If so, the k-fold
value you used should be reported.                                                                                                                       | As addressed in response 1, we did not use k-fold cross-validation during model training. Final model evaluation was performed using manually delineated reference data from UAV images that were completely excluded from the training process (see Section 2.1.1) which we clarify in the revised manuscript (Lines 403-407):                           |
|   |             |                                                                                                                                                                                                                                          | "We trained the segmentation model on citizen science plant photographs using a fixed data split, with 80% of the data for training and 20% for validation. The final segmentation model performance was evaluated using independent reference data derived from visual interpretation of UAV orthoimage transects, which were not used during training." |
| 4 | 286-
301 | The classification performance seems to be low for various species. Citizen science data helps reduce time and labor in reference data collection; however, we also need to make sure output data are accurate and usable. With this low | We acknowledge the reviewer's concerns regarding segmentation accuracy and appreciate the forward-looking suggestions. First of all, we would like to highlight that using citizen science data for drone-based remote sensing is still in its infancy, and we are just pioneering the                                                                    |

accuracy, what do authors suggest for future works? Should we incorporate some UAV-based high accuracy labelled data in the model together with citizen science data to improve classification accuracy? Also, the hyperparameter tuning seems not to be well-performed in your deep learning model training, I recommend conducting a more exhaustive tuning and trying different deep learning architecture to see if the classification results are improved

possibilities. This study is not about providing an operational technology, but rather about exploring methodological ways to harness citizen science data and its potential for drone-based mapping.

Here, we demonstrate this potential in a very complex scenario with several broadleaved tree species with similar leaf forms. Given this pioneering character and the complexity of the case study, we think that the results are groundbreaking and open up possibilities for a series of followup studies. Clearly, there are many aspects that can be improved and explored in greater depth. The discussion section, specifically the subsection "Segmentation performance" presents several avenues that might be explored in future research, including higher orthoimage resolution, other segmentation methods or harnessing the increasing growth of citizen science datasets.

In the revised manuscript, we made it clearer that this study is of a pioneering nature and focuses on method development rather than providing a ready-to-use solution. Accordingly, we revised the abstract and the introduction. Here are some examples:

"Here, we explore the potential of an automated workflow [...]" (Line 12-13)

"We demonstrate the potential of this approach [...]" (Line 147-157)

We applied several strategies to improve the segmentation accuracy across all tree species, including data augmentation, modifications of photograph backgrounds and scaling, hyperparameter tuning, and adjustments to model architectures. However, visual similarities among certain species led to trade-offs, improving accuracy for one species sometimes decreased it for others. Over several months, we conducted a thorough model ablation study, and the results presented here are the final outcome. The hyperparameter tuning is now described

|                                                                                                      |                                                                                                                                                                                                                                                                                                                                                                                                                                                                                                                                                                                                                                                                                                                                                                                                                                 | in more detail in the manuscript (see comment #1).                                                                                                                                                                                                                                                                                                                                                                                                                                                                                                                                                                                                                                                                                                                                                                                                                                                                                                                                                                                                                                                                                                                                                                                                                                                                                                                                                                                                                                                                                                                                                                                                                                                                                                                                                            |
|------------------------------------------------------------------------------------------------------|---------------------------------------------------------------------------------------------------------------------------------------------------------------------------------------------------------------------------------------------------------------------------------------------------------------------------------------------------------------------------------------------------------------------------------------------------------------------------------------------------------------------------------------------------------------------------------------------------------------------------------------------------------------------------------------------------------------------------------------------------------------------------------------------------------------------------------|---------------------------------------------------------------------------------------------------------------------------------------------------------------------------------------------------------------------------------------------------------------------------------------------------------------------------------------------------------------------------------------------------------------------------------------------------------------------------------------------------------------------------------------------------------------------------------------------------------------------------------------------------------------------------------------------------------------------------------------------------------------------------------------------------------------------------------------------------------------------------------------------------------------------------------------------------------------------------------------------------------------------------------------------------------------------------------------------------------------------------------------------------------------------------------------------------------------------------------------------------------------------------------------------------------------------------------------------------------------------------------------------------------------------------------------------------------------------------------------------------------------------------------------------------------------------------------------------------------------------------------------------------------------------------------------------------------------------------------------------------------------------------------------------------------------|
| low stu spa scir con dis you aut ma scir res inc the sea important revenue. Ma mu con (m ima res htt | we of the main reasons that cause w segmentation accuracy in this udy could be the difference in the latial resolutions between citizen ience photos and UAV images. The possible solution for this screpancy could be that during our segmentation model training, athors may want to anipulate/resample citizen ience photos to different solutions, including the 0.22 cm solution of the UAV image, and corporate features extracted from ese layers into the final gmentation prediction to help approve the final segmentation sults (see below paper with milar idea, note: this is not a viewer's paper).  artins et al., 2020. Exploring ultiscale object-based envolutional neural network multi-OCNN) for remote sensing lage classification at high spatial solution. tps://doi.org/10.1016/j.isprsjprs.2 10.08.004 | We agree that differences in spatial resolution and perspective could present a challenge for our transfer learning approach. In our current implementation, we do resample and rescale the citizen science photos to various resolutions during training (see Section 2.3), including resolutions similar to the UAV imagery as the reviewer suggested.  Achieving a perfect resolution match is difficult due to variability in ground-level photo distances, image quality, and variation in the drone-based imagery (e.g., due to differences in canopy height). We found that applying a generic scaling strategy sufficiently reduced the level of detail across all ground photographs to match that of the UAV imagery. While a multiscale architecture like the provided reference explicitly models these changes in scale, standard ("vanilla") architectures can still learn multiscale phenomena on the fly when sufficient variability is present. We agree with the reviewer that a more detailed exploration of resolution in the context of both image augmentation and model architecture are good focal areas for accuracy improvements in future work around UAV imagery segmentation, and have updated the discussion to reflect this: (Lines 493-497, 568-570):  "However, this diversity can also hamper model performance if imagery is not curated to match the downstream tasks, which prompted our removal of extremely close and extremely far images during training. Incorporating additional task-specific image adjustments, such as spatial re-sampling to the resolution of the UAV imagery Martins et al. (2020) should further improve performance"  "More complex architectures or methods, such as transformer or deeper CNN architectures, which integrate multi-scale |

|  | feature extraction and attention
mechanisms, offer promising alternatives
(Li et al., 2024)." |
|--|-----------------------------------------------------------------------------------------------------|
|  |                                                                                                     |
|  |                                                                                                     |

|            | Reviewer #                                                                                                                                                                                                                                                                                                                                                                                  | 2                                                                                                                                                                                                                                                                                                                                                                                                                                                                                                                                                                                                                                                                                                                                                                                                                                                                                                                                                                                                                                                                                                                                                                                                                                                                                                                                                   |
|------------|---------------------------------------------------------------------------------------------------------------------------------------------------------------------------------------------------------------------------------------------------------------------------------------------------------------------------------------------------------------------------------------------|-----------------------------------------------------------------------------------------------------------------------------------------------------------------------------------------------------------------------------------------------------------------------------------------------------------------------------------------------------------------------------------------------------------------------------------------------------------------------------------------------------------------------------------------------------------------------------------------------------------------------------------------------------------------------------------------------------------------------------------------------------------------------------------------------------------------------------------------------------------------------------------------------------------------------------------------------------------------------------------------------------------------------------------------------------------------------------------------------------------------------------------------------------------------------------------------------------------------------------------------------------------------------------------------------------------------------------------------------------|
| ID Line    | Comment                                                                                                                                                                                                                                                                                                                                                                                     | Response                                                                                                                                                                                                                                                                                                                                                                                                                                                                                                                                                                                                                                                                                                                                                                                                                                                                                                                                                                                                                                                                                                                                                                                                                                                                                                                                            |
| ID Line  1 | I would recommend the authors to add a workflow chart to help readers understand the various types of methods and data used for the study. There are several AI/ML models employed for various different data processing, including both photographs and UAV imagery. I found it hard to connect the different processing steps, and how different data streams and AI/ML methods are used. | Thank you for this feedback. We agree a workflow diagram will help clarify our multi-layered pipeline. Originally, Figure 3 was meant to serve this purpose. We have revised the figure to more clearly describe the workflow and moved it to the Introduction to improve its prominence (Lines 113-114):  Close science of the Control of the Control of the Introduction to improve its prominence (Lines 113-114):  Figure 1: Shamatis workflow of the regneratation make the promise model of the united the Control of the Introduction to the report Justice that the Control of the Introduction model of the Language were reported to make. The workflow of the Introduction text to ensure that the terminology aligns with all elements presented in the workflow diagram (Lines 125-138):  "To address these limitations, we present a novel workflow that transforms weakly labeled, crowd-sourced plant photographs into high-quality segmentation makes (Fig. 1). Our approach leverages the Segment Anything Model (SAM), a state-of-the-art foundation model designed for generic segmentation with Gradient-weighted Class Activation Mapping (Grad-CAM) (Selvaraju et al. 2017). First, we train a computer vision model for a simple species classifications, Grad-CAM highlights image regions that contribute most to species |
|            |                                                                                                                                                                                                                                                                                                                                                                                             | classification, which we use to guide point-
based prompts for SAM to generate                                                                                                                                                                                                                                                                                                                                                                                                                                                                                                                                                                                                                                                                                                                                                                                                                                                                                                                                                                                                                                                                                                                                                                                                                                                                   |

accurate segmentation masks. This enables an automated mask creation from images with only species-level labels, eliminating the need for manual pixel-wise annotation. Lastly, we enhance the transferability of these citizen science-based training data and its image features to the drone scale by exchanging the textures of the background class with common background samples from drone imagery."

2

Second, not much information is presented in the Results, barely enough to understand the performance of the model. The authors did quite significant work on processing and segmenting the photographs from iNaturealist and Pl@ntNet. However, results about these processing and segmentation are completely missed in the Results. I am nervous the presentation of Results is disconnected with the Methods. Recommend the authors to carefully tie them together, especially, how F1 score, confusion matrix was calculated. The authors mentioned independent transect validation data were identified from UAV imagery, but did not mention where and how those were produced, distribution across species and space etc. I think it is also useful to present the species maps across the experiment plots.

Thank you for this helpful feedback! We acknowledge that the original Results section was too brief and appreciate the reviewer's suggestions. We have revised the Results section to include the missing results around mask generation and species distribution maps, and to ensure stronger alignment with the Methods section. In the Methods section, we have also elaborated the description of the independent test data creation (transects). We have added the following information in the method section (Lines 408-417):

We evaluated the model performance of the segmentation model using the F1 score. The F1 score combines both Precision and Recall into a single measure, balancing false positives and false negatives (Eq. 1). The formulas used to compute Precision, Recall, and the F1 score are provided below:

$$\begin{aligned} & \operatorname{Precision} = \frac{TP}{TP + FP} \\ & \operatorname{Recall} = \frac{TP}{TP + FN} \end{aligned} \tag{1} \\ & F_1 = 2 \times \frac{\operatorname{Precision} \times \operatorname{Recall}}{\operatorname{Precision} + \operatorname{Recall}} \end{aligned}$$

In addition, we computed a confusion matrix for each class to reveal systematic confusion between species. We obtained the confusion matrix based on the predicted and reference segmentation

masks on a per-pixel basis. For each class, we counted the number of True Positives (TP), False Positives (FP), False Negatives (FN), and True Negatives (TN).

We have added the following information to the Results section, including a new figure (Fig. 4) illustrating the results of automatic mask generation using citizen science plant photographs from iNaturalist and Pl@ntNet (Lines 421-427):

"Across the ten tree species, the automated mask creation generated precise segmentation masks. These masks clearly delineated the target species, accurately capturing leaf contours, edges, and complex and even small morphological features such as small twigs, petioles, and branches (Fig. 4). Even in complex image scenarios and across the heterogeneous scene components, such as hands or other species, the masks consistently indicated the silhouettes of the target species."

Additionally, we included a map displaying the predictions on the monoculture plots (Fig. 6).

with several examples per species, showing the large variability in the images, including both leaf-level and whole-plantlevel photographs. From this, it should become evident that the model can extract features describing the canopy structure. Figure A2: Examples of citizen science photographs illustrating the predicted camera-to-object acquisition distances, demonstrating the accuracy and utility of our CNN-based re-gression model (Soliani et al., 2022). Rows represent increasing predicted distances, ranging from close-up leaf-level details to entire trees and broader landscape views, highlighting the variability in training data. Individual predicted distances are indicated below each image. 4 I wonder what features the authors The reviewer is correct that we used only used for segmentation? It is clear that RGB image data for both the SAM-based the authors used only RGB imagery, segmentation and the CNN model. We did but are other indices or not compute or incorporate any additional transformations incorporated in the spectral or textural indices, as SAM is SAM segmentation? primarily trained on RGB imagery and hence, any sophisticated feature engineering does not appear to be necessary. This is confirmed by the overall high quality of the output (see Fig. 1). In the revised manuscript, we emphasized that we used only RGB imagery as input (Lines 238-241). "Using Grad-CAM, we located the pixels that were important for the model to reveal the approximate location of the species within the image. Then, we

| The author mentioned that photos/masks from citizen science were 'zoomed' out when applied as training for UAV imagery. What's the resolution after that? Is it comparable to UAV resolution?  Since the original citizen science images had varying resolutions, applying fixed factors for zooming-out did not result in a uniform output resolution. However, our aim was to approximate the UAV resolution as closely as possible. We tested several factors for the zoom-out, and the selected value provided the best model performance. We have clarified this process in the revised manuscript (Lines 306-310):  "Specifically, we duplicated each photograph and zoomed out the plant foreground by 60 %. This approach ensures that our training dataset includes both the original and zoomed-out photographs. The value of 60 % was set heuristically, since an effective resolution of the citizen science photos is not |   |                                                                                                                                                           | sampled points from these image regions as input for the segmentation mask generation using SAM. Thereby, SAM was directly applied to the raw citizen science photographs."                                                                                                                                                                                                                                                                                                                                                                                                                                                                            |
|----------------------------------------------------------------------------------------------------------------------------------------------------------------------------------------------------------------------------------------------------------------------------------------------------------------------------------------------------------------------------------------------------------------------------------------------------------------------------------------------------------------------------------------------------------------------------------------------------------------------------------------------------------------------------------------------------------------------------------------------------------------------------------------------------------------------------------------------------------------------------------------------------------------------------------------|---|-----------------------------------------------------------------------------------------------------------------------------------------------------------|--------------------------------------------------------------------------------------------------------------------------------------------------------------------------------------------------------------------------------------------------------------------------------------------------------------------------------------------------------------------------------------------------------------------------------------------------------------------------------------------------------------------------------------------------------------------------------------------------------------------------------------------------------|
| avallable.                                                                                                                                                                                                                                                                                                                                                                                                                                                                                                                                                                                                                                                                                                                                                                                                                                                                                                                             | 5 | photos/masks from citizen science
were 'zoomed' out when applied as
training for UAV imagery. What's the
resolution after that? Is it comparable | had varying resolutions, applying fixed factors for zooming-out did not result in a uniform output resolution. However, our aim was to approximate the UAV resolution as closely as possible. We tested several factors for the zoom-out, and the selected value provided the best model performance. We have clarified this process in the revised manuscript (Lines 306-310):  "Specifically, we duplicated each photograph and zoomed out the plant foreground by 60%. This approach ensures that our training dataset includes both the original and zoomed-out photographs. The value of 60% was set heuristically, since an effective resolution |

---

## Author Response (AR2)

Salim Soltani, Chair of Sensor-based Geoinformatics (geosense), Faculty of Environment and Natural Resources, University of Freiburg, salim.soltani@geosense.uni-freiburg.de

To the Executive Editor and Reviewers of Biogeosciences

05.09.25

Ref. No.: egusphere-2025-662- "Automated mask generation in citizen science smartphone photos and their value for mapping plant species in drone imagery "

Dear Dr. Feldman and Reviewers,

Thank you very much for your decision on our manuscript and for your comments. We agree with the reviewer's suggestion and have included the F1 score from the citizen science—based validation split. We have now added this information to the Results section, which reads as follows:

(Lines 325-330)

"The U-Net segmentation model was first trained using the automatically derived segmentation masks in 80 epochs. The best model across these epochs, as selected from a validation split of the citizen science data, resulted in an F1 of 0.89 across all tree species and the background class. This model was then applied to the UAV imagery and corresponding reference data. The evaluation on the UAV-based reference data yielded varying F1 scores for the different tree species (Fig. 5 and 6)."

Sincerely,

Salim Soltani

(on behalf of the Co-authors, Lauren E. Gillespie, Moises Exposito-Alonso, Olga Ferlian, Nico Eisenhauer, Hannes Feilhauer, and Teja Kattenborn)